**METHOD**

# MetaBinner: a high-performance and stand-alone ensemble binning method to recover individual genomes from complex microbial communities

Ziye Wang[1,2], Pingqin Huang[3], Ronghui You[1], Fengzhu Sun[4] and Shanfeng Zhu[1,5,6,7,8*]

*Correspondence:
zhusf@fudan.edu.cn

[1] The Institute of Science and Technology for Brain-inspired Intelligence, Fudan University, Shanghai, China
[2] School of Mathematical Science, Fudan University, Shanghai, China
[3] School of Computer Science and Shanghai Key Lab of Intelligent Information Processing, Fudan University, Shanghai, China
[4] Department of Quantitative and Computational Biology, University of Southern California, Los Angeles, USA
[5] Shanghai Qi Zhi Institute, Shanghai, China
[6] Key Laboratory of Computational Neuroscience and Brain-Inspired Intelligence (Fudan University), Ministry of Education, Shanghai, China
[7] MOE Frontiers Center for Brain Science and Shanghai Institute of Artificial Intelligence Algorithms, Fudan University, Shanghai, China
[8] Zhangjiang Fudan International Innovation Center, Shanghai, China

## Abstract

Binning aims to recover microbial genomes from metagenomic data. For complex metagenomic communities, the available binning methods are far from satisfactory, which usually do not fully use different types of features and important biological knowledge. We developed a novel ensemble binner, MetaBinner, which generates component results with multiple types of features by k-means and uses single-copy gene information for initialization. It then employs a two-stage ensemble strategy based on single-copy genes to integrate the component results efficiently and effectively. Extensive experimental results on three large-scale simulated datasets and one real-world dataset demonstrate that MetaBinner outperforms the state-of-the-art binners significantly.

**Keywords:** Binning, Metagenomics, Metagenome

## Background

Metagenomics, the genomic analysis of microbial communities, provides a culture-independent way for exploring the unknown microbial organisms [1, 2]. Computational methods play an important role in metagenomic studies [3, 4]. Among these computational methods, contig binning aims to put the assembled genomic fragments, contigs, from the same genome into the same bin. The contigs from these bins are then reassembled to form metagenome-assembled genomes (MAGs). It is crucial for reconstructing MAGs from metagenomes for further analysis, such as identifying the uncultured bacterial species or viruses [5–7], associating viruses or bacterium with complex diseases [7–9] and exploring population diversity [10]. The quality of the MAGs generated by the binners will affect the results of these subsequent analyses. In this paper, we focus on the contig binning methods in general metagenomic data analysis.

Several binning methods have been widely used. CONCOCT [11] is a representative binner that groups all the contigs into genomic bins directly. CONCOCT combines a coverage vector and a tetra-mer frequency vector into one vector for each contig. It uses principal components analysis (PCA) for dimensionality reduction and Gaussian Mixture Model (GMM) for contig binning. MetaBAT 2 [12] is an efficient adaptive binning method that groups some of the contigs whose binning results are the most reliable at first (e.g., the longer contigs) and then gradually add the remaining contigs into the formed genomic bins. MaxBin [13, 14] multiplies the probabilities $P_{dist}$ and $P_{cov}$ that a sequence belongs to a bin based on the nucleotide frequency distance and coverage, respectively. A deep learning-based binner, VAMB [15], has recently been developed, which utilizes variational autoencoders (VAE) [16] to convert nucleotide information and coverage information for binning. VAMB then clusters the transformed data using an adaptive iterative medoid method.

Despite the extensive studies, none of the individual binners performs best in all the situations [17, 18]. Therefore, ensemble binning methods are developed to improve the binning performance. The ensemble binning methods can be divided into two categories: (1) the binners that integrate the binning results of other contig binners, such as DAS Tool [19], Binning_refiner [20], and MetaWRAP [21], and (2) the stand-alone binners that integrate multiple different component binning results within the ensemble binner, such as BMC3C [22]. DAS Tool [19] realizes genome reconstruction through a dereplication, aggregation, and scoring strategy. It calculates the scores of bins obtained by different binners with bacterial or archaeal reference single-copy genes (rSCG) [23, 24] and chooses the bins with the highest scores. Binning_refiner [20] merges results from multiple binning algorithms according to the shared contigs of two bins. It takes the sets of shared contigs with sufficient total length as refined bins. MetaWRAP [21] uses Binning_refiner [20] to generate hybrid bin sets and chooses the final bins with CheckM [25], which estimates bin quality based on single-copy genes (SCGs). UniteM (https://github.com/dparks1134/UniteM) is an ensemble binner developed based on CheckM [25] and DAS Tool [19]. Its "greedy" mode uses the SCGs from the bacteria and archaea domain in CheckM to estimate the bin quality and to determine the highest quality MAGs. In contrast, BMC3C [22] is independent of the results from other binners. It repeats k-means clustering multiple times with random initializations to obtain multiple component binning results using the same feature matrix (e.g., 50 times). Then, it transforms the index of the results into an affinity matrix. Finally, normalized cut [26] is used for binning. The ensemble methods usually achieve better performance than the individual methods [27].

Although many methods have been proposed to tackle the binning task, a few fundamental issues remain unresolved. Firstly, important biological knowledge such as SCGs has largely been ignored in the clustering process by most individual binners and BMC3C. Single-copy marker genes are the genes identified as a single copy in a large proportion (e.g., 97%) of the genomes within a specific phylum [25]. Due to this characteristic of the single-copy marker genes, they can be used for estimating the completeness and contamination of the microbial genomes recovered from the metagenomes [25], which enables to evaluate the binning performance without reference genomes and assist the binning process [28, 29]. Secondly, individual binners and

BMC3C lack diversity in terms of features. An individual binner usually uses the same features, and BMC3C integrates multiple binning results using the same features. However, various combinations of the features may help reconstruct the complex structure of the metagenomic datasets. The lack of diversities in features also weakens the effectiveness of other ensemble binners that depend on the results from the individual binners. Thirdly, the high-performance ensemble binner, MetaWRAP [21], can only integrate no more than three binning results simultaneously.

Here, we develop a novel ensemble contig binner, MetaBinner, independent of the results from other individual binners. MetaBinner first utilizes single-copy gene information for k-means initialization and uses different type of features for the k-means clustering method to generate different component binning results effectively. It then integrates the component binning results using an efficient two-stage ensemble strategy inspired by MetaWRAP [21] and UniteM "greedy" strategy (https://github.com/dparks1134/UniteM). The main contributions of MetaBinner are as follows: (1) MetaBinner uses a novel "partial seed" strategy for k-means initialization to utilize SCG information in the clustering process and obtain component binning results with high quality. (2) MetaBinner uses multiple different features and initializations for k-means clustering to obtain results with diversity for integration. (3) MetaBinner uses a novel effective and efficient two-stage ensemble strategy based on MetaWRAP and UniteM to select the bins with high completeness and low contamination as the final results (see "Methods" section for details).

We have validated the binning performance of MetaBinner using AMBER [27] and CheckM [25] on three large-scale multi-sample simulated datasets and a real-world dataset. Our experimental results show that MetaBinner outperforms the state-of-the-art binners, including the individual binners CONCOCT, MetaBAT, MaxBin, and VAMB, as well as the ensemble binners DAS Tool, MetaWRAP, and BMC3C. Specifically, in terms of the numbers of the near-complete bins (>90% completeness and < 5% contamination), MetaBinner increases by 75.9% and 32.5% on average for the simulated datasets compared to the best individual binner and the second-best ensemble binner, respectively.

## Results

### MetaBinner: a novel ensemble method for contig binning

MetaBinner has five major steps: (i) construct the feature representations of contigs with coverage and composition information, (ii) determine the number of bins, (iii) generate binning results with multiple features and initializations, (iv) split bins with high contamination according to the single-copy genes, (v) incorporate the component binning results with a two-stage efficient ensemble strategy. The general pipeline of MetaBinner is shown in Fig. 1. Detailed explanations of the steps are given in the "Methods" section.

In the following, we compare the performance of MetaBinner with other individual binners (CONCOCT [11], MetaBAT [12, 30], MaxBin [13, 14], VAMB [15]) and ensemble binners (BMC3C [22], MetaWRAP [21], and DAS Tool [19]). Then, we report our experimental results to show the necessity and effectiveness of multiple features and initializations. Finally, we report the running time of the binners on several datasets.

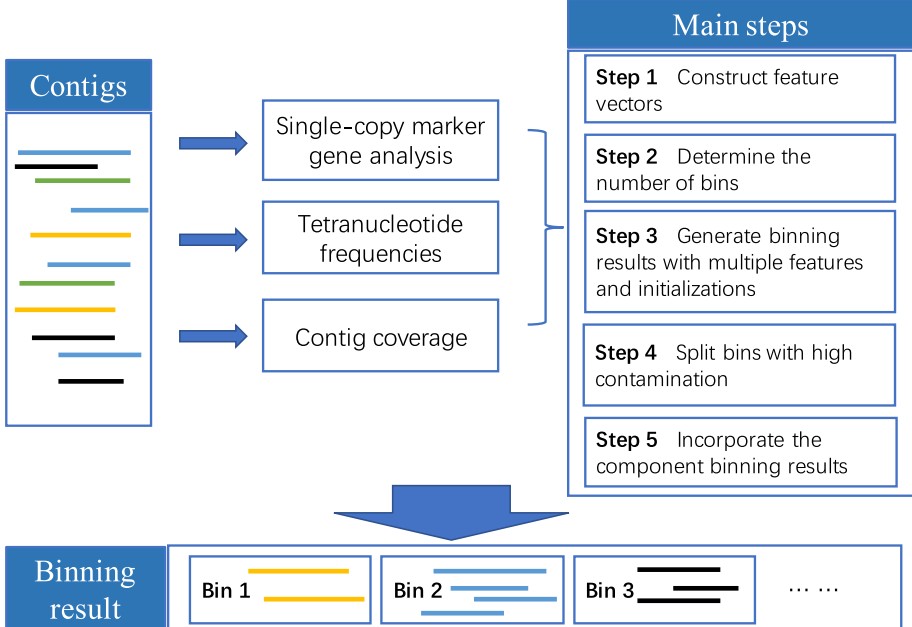

**Fig. 1** The general workflow of MetaBinner for contig binning

**MetaBinner outperforms other available contig binning methods on the simulated datasets evaluated by AMBER [27]**

We used three large-scale simulated multi-sample datasets to evaluate the binners using the evaluation metrics proposed in AMBER [27]. Detailed explanations of the datasets and evaluation metrics are given in the "Methods" section. Table 1 shows that MetaBinner can recover much more high-quality genomes than other binners under different completeness and contamination thresholds. None of the four individual binners perform best in all the three simulated datasets in terms of the number of high-quality bins, supporting the similar statement given in [17]. BMC3C automatically estimates the number of clusters and initializes it based on the number of contigs, which affects the stability of its performance. The total number of predicted bins per binner for each dataset are available in Additional file 1: Table S1. As shown in Additional file 1: Table S1 and Table 7, the numbers of predicted bins of BMC3C are quite different from those of other binners and the number of expected bins for the CAMI Airways dataset. For example, BMC3C generates 1560 bins for CAMI Airways, while the number of expected bins is 753.

Take the CAMI Gastrointestinal tract as an example for analysis. From the experimental results, we have three main findings. First, MetaBinner recovered the most high-quality genomes (>50% completeness and <10% contamination). Specifically, compared with the second-best binner, MetaBinner improves the numbers of near-complete (NC) genomes (>90% completeness and < 5% contamination; as defined in VAMB) from 112 to 147. Second, BMC3C, MetaBAT and CONCOCT assign the most base pairs at the cost of a lower Adjusted Rand Index (ARI) (Fig. 2a). Among the binners with the highest ARI (over 90%), MetaBinner assigns the most base pairs. Third, among all the binners, MetaBinner achieves the highest average completeness of all predicted bins. Its average purity is close to those of the binners with the highest average purity (Fig. 2b).

**Table 1** Performance comparison of the binners on the simulated datasets evaluated by AMBER

| Dataset | Methods | Metrics | | | | | |
|---|---|---|---|---|---|---|---|
| | | #bins (>50% comp <10% cont) | #bins (>70% comp <10% cont) | #bins (>90% comp <10% cont) | #bins (>50% comp <5% cont) | #bins (>70% comp <5% cont) | #bins (>90% comp <5% cont) |
| CAMI Airways | CONCOCT | 42 | 38 | 32 | 42 | 38 | 32 |
| | MaxBin | 88 | *82* | *64* | 67 | 62 | *51* |
| | MetaBAT | *92* | 80 | 56 | 84 | *74* | *51* |
| | VAMB | 90 | 75 | 47 | *86* | 72 | 46 |
| | BMC3C | 18 | 8 | 6 | 13 | 6 | 4 |
| | DAS Tool | 106 | 101 | 80 | 94 | 91 | 71 |
| | MetaWRAP | 136 | 119 | 83 | 131 | 118 | 82 |
| | MetaBinner | **215** | **191** | **144** | **186** | **169** | **129** |
| CAMI Gastrointestinal tract | CONCOCT | 66 | 64 | 61 | 62 | 60 | 57 |
| | MaxBin | *114* | *110* | *101* | *106* | *103* | *97* |
| | MetaBAT | 97 | 93 | 83 | 93 | 89 | 80 |
| | VAMB | 57 | 55 | 42 | 56 | 55 | 42 |
| | BMC3C | 12 | 9 | 7 | 7 | 4 | 3 |
| | DAS Tool | 130 | 125 | 116 | 124 | 120 | 111 |
| | MetaWRAP | 134 | 126 | 114 | 131 | 123 | 112 |
| | MetaBinner | **183** | **173** | **152** | **176** | **166** | **147** |
| CAMI mouse gut | CONCOCT | 102 | 102 | 88 | 92 | 92 | 79 |
| | MaxBin | *449* | *435* | *351* | *423* | *409* | *332* |
| | MetaBAT | 375 | 350 | 286 | 361 | 338 | 277 |
| | VAMB | 382 | 370 | 293 | 372 | 363 | 288 |
| | BMC3C | 315 | 306 | 262 | 297 | 290 | 253 |
| | DAS Tool | 469 | 461 | 377 | 441 | 435 | 357 |
| | MetaWRAP | 506 | 487 | 377 | 500 | 482 | 375 |
| | MetaBinner | **575** | **532** | **421** | **535** | **503** | **409** |

The best results among all the methods are in bold, while the best results among the individual binners are italicized. The input binning results of MetaWRAP and DAS Tool are generated by CONCOCT, MaxBin, and MetaBAT. "#bins (>50% comp <10% cont)" denotes the number of recovered bins that have >50% completeness and <10% contamination

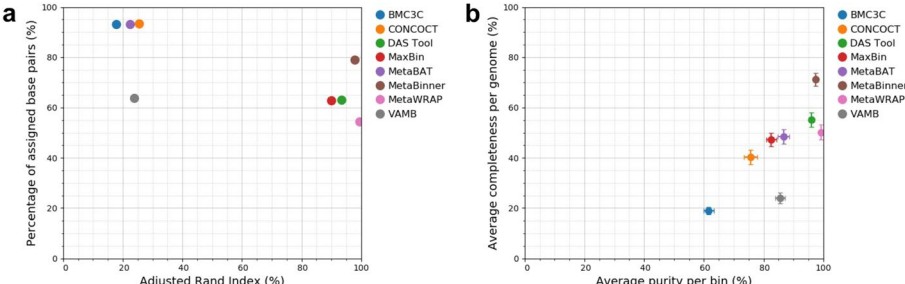

**Fig. 2** Assessing binners on the CAMI Gastrointestinal tract dataset based on base pairs. **a** Adjusted Rand index (*x*-axis) versus percentage of assigned base pairs (*y*-axis). **b** Average purity (*x*-axis) versus average completeness (*y*-axis) of all predicted bins per binner. The figures are obtained via AMBER [27]

### MetaBinner produces the most near-complete MAGs on the real dataset

For the real dataset, the true genomes in the metagenomes are unknown. In this situation, CheckM is widely used in studies [31, 32] for selecting the high-quality bins from the Metagenome-Assembled Genomes (MAGs). As shown in [4], the results of CheckM and AMBER are largely consistent on the CAMI mouse gut dataset. As shown in Table 2, MetaBinner and MetaWRAP achieve the best overall performance. Among the individual binners, VAMB achieves the best performance. MetaBinner can recover the most near-complete genomes. The assembly results of the sequencing data may affect the follow-up binning performance. Therefore, we assembled the reads using another popular assembler, metaSPAdes [33], and used the assembled contigs for binning. The results are given in Additional file 1: Table S2, and MetaBinner also achieves the best performance.

It is possible that using CheckM to evaluate the resulting bins for the real dataset would be biased toward MetaBinner (using domain-specific marker sets of CheckM for estimating bin scores) and MetaWRAP (using CheckM directly). Therefore, we explored another approach to evaluate the binning performance for the real dataset. First, we obtained the species-level annotations of the contigs by aligning them to the NCBI's nt database using TAXAassign v0.4 (https://github.com/umerijaz/taxaassign) as done in [11, 34]. For the STEC (MEGAHIT assembly) dataset, a total of 46,609 out of 255,484 contigs were labeled on the species level for evaluation. Then, we used AMBER [27] to evaluate the binners based on the contigs labeled on the species level. Note that we did not re-run the binning methods but extracted labeled contigs from the results of each binner for evaluation. As shown in Additional file 1: Table S3, MetaBinner still performs the best in producing the most MAGs with the highest completeness (>90% completeness and <5% or <10% contamination). For example, MetaBinner produces 24 most near-complete MAGs (>90% completeness and <5% contamination), which is followed by VAMB and MaxBin (19 near-complete MAGs). We also find that VAMB and BMC3C perform well in producing middle-complete and less-complete MAGs (>70% or >50%

**Table 2** Performance comparison of the binners on the real dataset evaluated by CheckM

| Dataset | Methods | Metrics | | | | | |
|---|---|---|---|---|---|---|---|
| | | #bins (>50% comp <10% cont) | #bins (>70% comp <10% cont) | #bins (>90% comp <10% cont) | #bins (>50% comp <5% cont) | #bins (>70% comp <5% cont) | #bins (>90% comp <5% cont) |
| STEC (MEGAHIT assembly) | CONCOCT | 95 | 63 | 26 | 78 | 48 | 19 |
| | MaxBin | 106 | 76 | *41* | 64 | 42 | 23 |
| | MetaBAT | 101 | 60 | 24 | 92 | 55 | 22 |
| | VAMB | *145* | *88* | 37 | *116* | *65* | *29* |
| | BMC3C | 120 | 61 | 23 | 106 | 53 | 20 |
| | DAS Tool | 99 | 71 | 38 | 78 | 51 | 26 |
| | MetaWRAP | 155 | 96 | 33 | **139** | **81** | 29 |
| | MetaBinner | **164** | **105** | **48** | 122 | 72 | **38** |

The best results among all the methods are in bold, while the best results among the individual binners are italicized. The input binning results of MetaWRAP and DAS Tool are generated by CONCOCT, MaxBin, and MetaBAT. "#bins (>50% comp <10% cont)" denotes that the number of recovered bins that have >50% completeness and <10% contamination

completeness). These suggest that VAMB and BMC3C generate bins with low contamination but low completeness. As shown in Additional file 1: Table S1, the total number of predicted bins of BMC3C for STEC (MEGAHIT assembly) is much larger than other binners, which means that BMC3C may produce more bins with low contamination and low completeness. On the other hand, VAMB clustered the contigs into 75,526 bins first and then kept the 256 bins more than 200,000 bp as the predicted bins. This means that VAMB tends to generate bins with high purity but less completeness, which is consistent with the result of CAMI II [18]. All these results highlight the advantage of MetaBinner in producing the most near-complete MAGs on the real dataset.

### The effect of generating component binning results with multiple features and initializations

When running the k-means++-based method for comparison, we use the length of each contig to set the weight for each contig. We take the binning results of CAMI Airways as the example.

1) The effect of the "partial seed" method.

To demonstrate the effect of the "partial seed" method, we ran k-means++ randomly for three times and compared the results with three "partial seed" binning results for each feature matrix generated by "Step 3". We suppose that the improvement of binning results is due to two reasons: we used contigs containing single-copy marker genes as cluster centers and added the regular k-means++ initialization part. To further prove this point, we also compared the "partial seed" results with the results using the same cluster centers but without the regular k-means++ initialization part. "Seed k-means" indicates that we ran k-means++ using the features of the contigs containing the chosen single-copy marker gene as the cluster centers, and the bin number is the same as the number of corresponding contigs.

From the experimental results given in Table 3, we have the following main findings. Firstly, the component binning results generated using $X_{combo}$ feature matrix (109 high-quality bins) have the best quality compared with those using other feature matrices (41 and 93 high-quality bins). Secondly, the length-weight strategy can improve the binning performance. Thirdly, using the contigs containing the single-copy marker genes as the cluster centers can improve the binning performance. Take the results from using $X_{combo}$ feature matrix as an example. The "seed k-means" method recovers about 100 high-quality bins on average, compared with 75.33 generated by regular k-means++. Finally, the component with regular k-means++ initialization in "partial seed" helps in binning, especially for the component binning results generated using $X_{combo}$ and $X_{cov}$ feature matrix.

2) The effect of incorporating binning results using three kinds of feature combinations.

We use the changes of the final output of MetaBinner to measure the effect. MetabinA, MetabinB, and MetabinC denote the ensemble results of the component binning results generated using $X_{combo}$, $X_{cov}$, and $X_{com}$, respectively. MetabinAB denotes the MetaBinner results after removing the parts related to MetabinC during the second step of integration (see Fig. 3). The results drop from 215 to 203, 197, and 189 after removing the components using each feature matrix in terms of the number of high-quality bins (Table 4). Interestingly, the MetabinA (175 high-quality bins) even has better

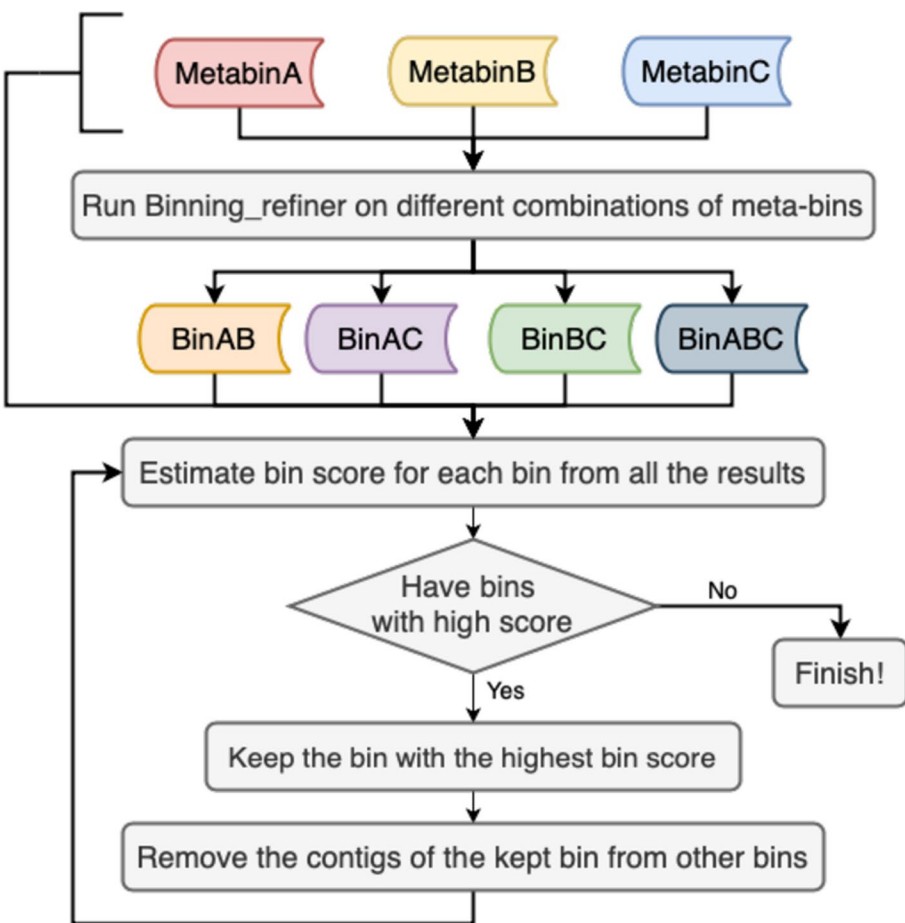

**Fig. 3** The ensemble strategy workflow (the second stage). BinAB denotes the refined binning results of MetabinA and MetabinB using Binning_refiner [20]. The quality of a bin is described by a bin score, which is computed as 100 × (completeness -3 × contamination). If a bin has a bin score higher than 10, low contamination (<15%), and high completeness (>50%), we regard it as a bin with a high score

performance than the second-best method (MetaWRAP: 136 high-quality bins) for this dataset (Table 4).

3) The effect of integrating component results using the proposed ensemble strategy instead of DAS Tool.

MetabinA (DAS Tool), MetabinB (DAS Tool), and MetabinC (DAS Tool) denote the DAS Tool integration results of the component binning results generated using $X_{combo}$, $X_{cov}$, and $X_{com}$, respectively. Table 5 shows the performance comparison of MetaBinner and DAS Tool using one feature combination in CAMI Airways dataset. The proposed ensemble strategy has better performance than DAS Tool on all three feature combinations. The number of high-quality bins using $X_{combo}$ feature matrix improves from 147 to 175.

### Running time of the binners

All the results given in this section were run on two Intel Xeon CPUs (E5-2660 v3, 2.60GHz) with 128G RAM. We ran all the binners with multiple threads. Table 6 shows the running time of MetaBinner, the three individual binners for DAS Tool

**Table 3** Performance comparison of "partial seed" and regular k-means++ in terms of recovered high-quality bins in CAMI Airways dataset

| Feature | Methods | Metrics | | | | | |
|---|---|---|---|---|---|---|---|
| | | #bins (>50% comp <10% cont) | #bins (>70% comp <10% cont) | #bins (>90% comp <10% cont) | #bins (>50% comp <5% cont) | #bins (>70% comp <5% cont) | #bins (>90% comp <5% cont) |
| | k-means++ average (no length weighting) | 32.67 | 26.33 | 9.33 | 27.67 | 21.67 | 6.67 |
| $X_{combo}$ | k-means++ average | 75.33 | 60.33 | 39.00 | 65.67 | 52.67 | 35.33 |
| | seed k-means average | 100.33 | 91.33 | 63.00 | 74.67 | 67.33 | 49.00 |
| | partial seed average | **109.33** | **96.00** | **65.67** | **81.67** | **71.33** | **50.67** |
| | k-means++ average (no length weighting) | 30.67 | 26.67 | 20.33 | 18.00 | 16.00 | 11.33 |
| $X_{cov}$ | k-means++ average | 57.00 | 51.00 | 45.33 | 43.67 | 38.67 | 35.00 |
| | seed k-means average | 89.00 | 82.67 | 75.00 | 66.00 | 61.33 | 57.00 |
| | partial seed average | **93.00** | **86.33** | **78.67** | **71.00** | **66.33** | **61.00** |
| | k-means++ average (no length weighting) | 14.33 | 7.33 | 1.00 | 11.33 | 7.00 | 1.00 |
| $X_{com}$ | k-mean++ average | 30.33 | 22.67 | 13.33 | 21.00 | 16.00 | 7.33 |
| | seed k-means average | 40.67 | **29.67** | 15.67 | **24.00** | 16.00 | 7.67 |
| | partial seed average | **41.00** | 29.00 | **16.67** | 23.67 | **16.33** | **8.67** |

The best results based on each feature matrix are in bold. $X_{combo}$ denotes the feature matrix combining coverage and composition information, $X_{cov}$ denotes the feature matrix using coverage information, and $X_{com}$ denotes the feature matrix using composition information (see the "Methods" section for more details). "#bins (>50% comp <10% cont)" denotes that the number of recovered bins that have >50% completeness and <10% contamination. "no length weighting" denotes that all contigs are assigned equal weight while running k-means++

and MetaWRAP, and the two ensemble binners on different datasets (CAMI Airways and CAMI mouse gut). Since most binners can share the steps of generating composition and coverage files, we only compared the running time required after generating these files. For the dataset with more samples (CAMI mouse gut), it takes much more time to run MaxBin (more than 7100 min) compared with other binners, so the whole running time of MetaBinner (about 1514 min) is much less than other ensemble binners. For the CAMI airways dataset, the running time of MetaBinner is close to that of DAS Tool.

**Table 4** Performance comparison of MetaBinner and the results of MetaBinner after removing binning results using one or two kinds of feature combinations in CAMI Airways dataset

| Methods | Metrics | | | | | |
|---|---|---|---|---|---|---|
| | #bins (>50% comp <10% cont) | #bins (>70% comp <10% cont) | #bins (>90% comp <10% cont) | #bins (>50% comp <5% cont) | #bins (>70% comp <5% cont) | #bins (>90% comp <5% cont) |
| MetabinA | 175 | 157 | 113 | 130 | 120 | 87 |
| MetabinB | 145 | 139 | 116 | 113 | 110 | 89 |
| MetabinC | 148 | 119 | 96 | 114 | 92 | 73 |
| MetabinAB | 203 | 184 | 134 | 163 | 151 | 111 |
| MetabinAC | 197 | 173 | 130 | 167 | 151 | 116 |
| MetabinBC | 189 | 166 | 135 | 163 | 146 | 118 |
| MetabinABC (MetaBinner) | **215** | **191** | **144** | **186** | **169** | **129** |

The best results are in bold. "#bins (>50% comp <10% cont)" denotes the number of recovered bins that have >50% completeness and <10% contamination

**Table 5** Performance comparison of the results of MetaBinner and DAS Tool using one feature combination in CAMI Airways dataset

| Methods | Metrics | | | | | |
|---|---|---|---|---|---|---|
| | #bins (>50% comp <10% cont) | #bins (>70% comp <10% cont) | #bins (>90% comp <10% cont) | #bins (>50% comp <5% cont) | #bins (>70% comp <5% cont) | #bins (>90% comp <5% cont) |
| MetabinA | **175** | **157** | **113** | **130** | **120** | **87** |
| MetabinA (DAS Tool) | 147 | 138 | 106 | 108 | 103 | 80 |
| MetabinB | **145** | **139** | **116** | **113** | **110** | **89** |
| MetabinB (DAS Tool) | 131 | 127 | 112 | 100 | 98 | 84 |
| MetabinC | **148** | **119** | **96** | **114** | **92** | **73** |
| MetabinC (DAS Tool) | 112 | 106 | 93 | 86 | 81 | 71 |

The best results based on each feature matrix are in bold. "#bins (>50% comp <10% cont)" denotes the number of recovered bins that have >50% completeness and <10% contamination

**Table 6** The running time of the binners

| Binners | CAMI Airways | CAMI mouse gut |
|---|---|---|
| CONCOCT | 39m | 167m |
| MaxBin | 934m | 7121m |
| MetaBAT | **23m** | **16m** |
| DAS Tool | 20m | 26m |
| MetaWRAP | 547m | 1601m |
| DAS Tool (+running time of three individual binners) | **1016m** | 7320m |
| MetaWRAP (+running time of three individual binners) | 1543m | 8905m |
| MetaBinner | 1058m | **1514m** |

The best results among the individual binners and the ensemble binners are in bold

## Discussion

In this paper, we introduced MetaBinner, a novel stand-alone ensemble binner for large-scale contig binning. Firstly, MetaBinner generates binning results mainly using "partial seed" k-means with multiple types of features and initializations. Then, MetaBinner applies an effective and efficient two-stage ensemble strategy to integrate the component binning results. We compared MetaBinner with advanced binning tools, CONCOCT, MaxBin, MetaBAT, VAMB, BMC3C, DAS Tool, and MetaWRAP on four datasets, and MetaBinner has the best overall performance among all the datasets.

One key idea of MetaBinner is to combine single-copy marker gene information and k-means for generating high-quality and diverse component binning results efficiently. Specifically, MetaBinner employs the "partial seed" strategy to reduce the impact of inaccurate estimation of the bin number, which may be caused by the contigs from other categories of the taxon in complex metagenomic communities or the imperfect metagenomics assembly [4]. Furthermore, MetaBinner improves the binning performance by generating component results using different feature matrices and integrating the component results with a two-stage ensemble strategy based on SCGs. The above points make up for the shortcomings of the existing binning methods. We report the respective effect of the "partial seed" strategy and the effect of multiple features and initializations in the "Results" section. In addition, as a stand-alone ensemble binner, MetaBinner is efficient and does not utilize the results from other individual binners as the other two popular ensemble binners, MetaWRAP and DAS Tool. Finally, the results of MetaBinner can be integrated into different ensemble approaches (such as MetaWRAP and DAS Tool) as a component to achieve better performance. Other individual binners can also be integrated into MetaBinner flexibly by replacing MetabinA, B, or C with their results (see Fig. 3).

MetaBiner achieves the best overall performance on the real dataset (Tables 2 and S3), especially using CheckM for evaluation. On the other hand, VAMB and BMC3C performed pretty well based on AMBER evaluation for the contigs with species-level annotation (Table S3). Here we propose a possible explanation. Only the annotated contigs can be used for AMBER evaluation, but the annotated contigs of each species cannot cover the entire genome. Therefore, the evaluation methods may be biased towards the binners that generate bins with low contamination but low completeness. To prove our hypothesis to a certain extent, we regarded the annotated contigs of each species as a putative bin. We then evaluated the contamination and completeness of these putative bins using CheckM. Among the 256 putative bins, 252 bins have <10% contamination, but only 39 of the 252 low-contamination bins have >50% completeness. It is still challenging to evaluate the contamination and completeness of the genomes from the real metagenomes without bias.

The Critical Assessment of Metagenome Interpretation (CAMI) provides a general standard for comparing several important metagenomic computational methods, including contig binning [3, 18]. According to the results reported in their benchmark manuscript [18], the earlier version of MetaBinner achieved the best overall performances on two of the three simulated benchmark datasets (marine and strain madness datasets) used in CAMI II challenges. As for the third dataset, the

plant-associated dataset, MetaBinner recovered the most high-quality genomes from its gold-standard assemblies and hybrid assemblies.

Despite the successes of the MetaBinner for large-scale contig binning, it still has limitations. For example, the single-copy gene sets only contain bacterial and archaeal reference genes. In the future, we would like to explore the way of integrating the marker genes from other taxa, such as microbial eukaryotes [35], into the binning pipeline to resolve more complex microbial communities. Furthermore, since some component binning results for integration are generated using coverage information alone as features, we recommend applying MetaBinner to multi-sample datasets.

## Conclusions

MetaBinner is a powerful binning method for recovering individual genomes from complex microbial communities. The experimental results on real and simulated datasets show that MetaBinner outperforms the cutting-edge individual and ensemble binners. It will be useful to the field in analyzing metagenomic sequencing data.

## Methods

In this section, we present the following: (1) the descriptions of the benchmark datasets, (2) the details of each step in MetaBinner, (3) the metrics to evaluate the binning performance, and (4) the implementation and parameter settings of different methods.

### Datasets

#### The simulated datasets

We used one benchmark dataset, CAMI mouse gut, from the recent CAMI benchmarking toolkit tutorial [4] and two other "toy" human short-read datasets from CAMI II Challenge [18], CAMI Airways and CAMI Gastrointestinal tract, to evaluate the performance of the binners. All the simulated datasets were downloaded from the CAMI portal datasets at https://data.cami-challenge.org/. Most competing methods can only cluster the contigs longer than 1000 bp. Therefore, we kept the contigs of the gold standard cross-sample assembly longer than 1000 bp for binning. More details about the ground truth annotations of the contigs are given in Additional file 1. We used the simulated Illumina HiSeq reads that CAMI provided to generate the coverage information. Table 7 shows the general information of the simulated datasets.

**Table 7** Datasets used in the experiments

| Data Type | Dataset | # samples | # contigs (> 1000bp) | # genomes | # excepted bins |
|---|---|---|---|---|---|
| Simulated | CAMI Airways | 10 | 285047 | 828 | 753 |
| | CAMI Gastrointestinal tract | 10 | 57088 | 259 | 246 |
| | CAMI Mouse gut | 64 | 241451 | 791 | 769 |
| Real | STEC (MEGAHIT assembly) | 53 | 255484 | ... | ... |
| | STEC (metaSPAdes assembly) | 53 | 226544 | ... | ... |

"# genomes" denotes the number of genomes used for simulating the datasets. "# excepted bins" denotes the number of true bins expected among the contigs (> 1000bp)

### The real dataset

To assess the performance of the binners on large real datasets, we used a real dataset with multiple samples, the "STEC" dataset. The "STEC" dataset [36] contains 53 samples from a set of fecal specimens in the PRJEB1775 study (https://www.ebi.ac.uk). MetaWRAP [21] is a modular pipeline, and its "Assembly" module allows users to assemble metagenomic reads with metaSPAdes [33] or MEGAHIT [37]. The reads from all the samples are co-assembled by MetaWRAP-Assembly module with default parameters and the default assembler (MEGAHIT). The binning results based on the MEGAHIT assembly are shown in the "Results" section. The quality of the assemblies may affect the follow-up binning performance. Therefore, we also used the MetaW-RAP-Assembly module and the metaSPAdes assembler to co-assemble the reads from all the samples. The binning results based on this assembly are given in Additional file 1: Table S2. The general information of the real dataset is given in Table 7.

### The MetaBinner algorithm

Figure 1 shows the framework of MetaBinner, which consists of two modules: (1) "Component module" includes steps 1–4, developed for generating high-quality, diverse component binning results, and (2) "Ensemble module" includes step 5, developed for recovering individual genomes from the component binning results. More descriptions of each step are as follows.

### *Step 1: Construct feature vectors for metagenomic contigs*

Each contig co-assembled from $M$ samples can be represented with the combination of a coverage vector ($M$ dimensional) and a composition vector ($T$ dimensional) as done in previous studies [34, 38], where $T$ is the number of distinct tetramers. The coverage vector and the composition vector denote the coverage profiles across the $M$ samples and the tetramer frequency, respectively. A small value is added to each entry of the vectors (0.01 for the coverage vector; one for the composition vector) to handle zero values. Then the coverage matrix and the composition matrix are normalized as in COCACOLA [34]. For some datasets with a dozen of high-quality sequencing samples, using coverage vector only could yield good binning results. In such case, each contig can be represented by the $M$ dimensional coverage vector only. Furthermore, different organisms usually have different tetra-mer composition profiles [39, 40]. Therefore, the feature matrix of the contigs is denoted as $X_{combo} \in \mathbb{R}^{N \times (M+T)}$, $X_{cov} \in \mathbb{R}^{N \times M}$ or $X_{com} \in \mathbb{R}^{N \times T}$, where $N$ denotes the number of contigs. We did log transformation for each feature matrix as done in CONCOCT for the $X_{combo}$ feature matrix. In this way, we obtained three feature matrices for each dataset.

### *Step 2: Determine the number of bins*

Similar to SolidBin [38] and COCACOLA [34], we utilized the set of single-copy genes universal for bacteria and archaea provided by [13] to estimate the number of genomes in the metagenomic data. As stated in [13], some marker genes may be fragmented into pieces, influencing the estimation of the bin number. So we calculated the number of contigs containing each marker gene, and then used the third quartile

value of the numbers in ascending order to determine the initial bin number $k_0$. A list of numbers larger than $k_0$ was then sequentially tried as the bin numbers in the k-means algorithm (see Additional file 1: Fig. S1). The bin number yielding the largest silhouette coefficient [41] value of the binning result is chosen as the final bin number.

***Step 3: Generate binning results with multiple features and initializations.***

To generate high-quality and diverse component binning results, we proposed the "Partial Seed" strategy based on k-means++ [42], a variant of k-means. More descriptions about k-means++ are given in Additional file 1. Let $K$ denote the bin number estimated by step 2. Instead of randomly choosing the $K$ cluster centers, we used SCG information to define cluster centers. For a particular SCG, suppose that there are $l$ ($l < K$) contigs containing this gene. These $l$ contigs should belong to different genomes, which were used to initialize $l$ cluster centers. Then we used k-means++ to generate the other $K - l$ initial clustering centers and get the binning results.

To achieve a better integration effect while using the ensemble module, we can produce diverse binning results using different sets of fixed initial clustering centers. Therefore, we kept the first, second, and third quartile values of the numbers of contigs containing each marker gene. Similar to MaxBin, the shortest marker gene corresponding to each number was selected. In this way, we obtained three sets of designated initial clustering centers for each feature matrix.

To generate the binning results without considering SCG information for integration, we also ran regular k-means++ on each feature matrix. In summary, four binning results are generated for each feature matrix. Three of them are from the "partial seed" method, and one is from regular k-means++. Since there are three feature matrices ($X_{combo}$, $X_{cov}$, and $X_{com}$) for each dataset, twelve binning results are generated for the next step.

Similar to MetaBinner, MaxBin [13, 14] also applies SCG information for initializing the parameters of the clustering algorithm. However, in MaxBin, the bin number equals the number of the contigs containing the certain SCG. Note that the single-copy gene sets do not cover all the genomes in the microbial communities. In this way, some contigs from the genomes without the certain SCG may be assigned into the wrong bins, resulting in high contamination.

***Step 4: Split bins with high contamination according to the single-copy genes.***

To generate more bins with low contamination for further integration, we did post-processing for each binning result produced in step 3. We ran CheckM for one binning result and obtained the contigs having the single-copy genes. Similar to UniteM, we then used the information to estimate each bin's contamination and completeness of each component binning result using the scoring strategy in CheckM. BinSanity [29] applies a composition-based refinement to handle the highly contaminated or low-completion bins. Here, we only handle the highly contaminated bins. In our paper, if a bin has high contamination ($>= 50\%$) and completeness ($>= 70\%$), we split it by estimating the number of sub-bins of the bin using the same approach as in step 2 and running the k-means++ clustering. More details about the selection of the parameters are given in Additional file 1. We regard the binning results generated by Step 4 as "component binning results".

***Step 5: Incorporate the component binning results with an ensemble module.***

To incorporate the component binning results efficiently and effectively, we completed the integration process in two stages. The quality of binning results obtained by different input matrices may be markedly different. In the first stage of the ensemble module, we separately integrated four component binning results generated by Step 4 for each input feature matrix. The quality of a bin is described by a bin score, which is computed as $100 \times$ (completeness $-3 \times$ contamination). First, the bins with high bin scores estimated by the SCGs for bacterial and archaeal genomes will be selected. Then, the contigs in the selected bins will be removed from other bins. The above two operations are repeated until there are no high-quality bins. In the second stage, we integrated the three ensemble results of the first stage, as shown in Fig. 3. First, Binning_refiner is applied directly to refine bins generated from the first stage to produce bins with low contamination levels as done in MetaWRAP. Then, the same method as the first stage is used for selecting high-quality bins.

The process of picking high-quality bins for the two stages is highly similar to UniteM's greedy strategy. The main difference is that we only ran CheckM once for each domain (bacteria and archaea domains) to get the SCG information for all the contigs in step 4, instead of running it for all the component results as done in UniteM. For the same reason, the second stage does not need to run CheckM as many times as MetaWRAP. Our two-stage strategy can integrate the twelve component binning results efficiently by avoiding running CheckM on multiple component binning results, which is quite time-consuming. The differences between our ensemble strategy and other ensemble binners are summarized in Table 8.

## Evaluation metrics

For the simulated datasets, we used AMBER [27], which implements the metrics in the first CAMI binning challenge for evaluation [3, 4]. AMBER metrics are calculated based on a gold standard mapping result of the contigs or reads, so it is only suitable for the simulated datasets. We used the following quantities to evaluate the binning results: (a) number of high-quality genomes, (b) adjusted Rand index ($x$-axis) versus percentage of assigned base pairs ($y$-axis), and (c) average purity ($x$-axis) versus average completeness ($y$-axis) of all predicted bins per method. The definitions of the

**Table 8** The comparison between MetaBinner and other ensemble binners in ensemble strategy

|  | BMC3C | DAS Tool | UniteM | MetaWRAP | MetaBinner |
|---|---|---|---|---|---|
| 1. Independent of other individual binners | ✓ | × | × | × | ✓ |
| 2. Use single-copy gene (SCG) information | × | ✓ | ✓ | ✓ | ✓ |
| 3. Run CheckM on only one of all component binning results | – | – | × | × | ✓ |
| 4. Use Binning_refiner to generate some of the candidate bins | × | × | × | ✓ | ✓ |
| 5. Can directly integrate more than three component binning results | – | ✓ | ✓ | × | ✓ |

"✓" denotes that the binner conforms to the statement given in the first column. "×" denotes that the binner does not conform to the statement given in the first column. "–" denotes "not applicable"

criteria are given in AMBER [27]. The high-quality genomes are defined as genomes with > 50% completeness and < 10% contamination as done in [4].

For the real dataset without known genome assignments, we applied CheckM [25] for evaluation to obtain the bin's completeness and contamination scores.

### Implementation and parameter settings

We compared MetaBinner with seven advanced binners: CONCOCT-1.0.0, MaxBin 2.2.6, MetaBAT 2.12.1, VAMB 3.0.2, MetaWRAP 1.2.1, DAS Tool 1.1.2, and BMC3C, respectively. MetaWRAP and DAS Tool need to integrate the results from other binners. CONCOCT, MaxBin, and MetaBAT were chosen for these two ensemble binners as done in MetaWRAP [21]. We ran CONCOCT-1.0.0, MaxBin 2.2.6, and MetaBAT 2.12.1 using the binning module of MetaWRAP with the "−universal" parameter to use universal marker genes, which can improve binning for the Archaea genomes. We ran MetaWRAP with "-c 50" to set the minimum % completion of the bins. The coverage profiles of the contigs were obtained via a script of MetaWRAP 1.2.1, "binning.sh", while running the MetaWRAP's binning module. This script alig176ns the reads against the contigs using BWA [43] and calculates the mean of the base depths (coverage) for each contig. For a fair comparison with other methods, we ran coassembly mode of VAMB with "−jgi depth.txt −minfasta 200000". The results of the simulated and real datasets were evaluated by AMBER 2.0.21-beta and CheckM v1.1.3.

MetaWRAP is a modular pipeline, and we regard its "bin_refinement" module as MetaWRAP in this paper if there is no additional explanation. The detailed commands for executing the compared binning methods and assemblers are given in Additional file 1.

### Supplementary Information

---

Additional file 1. Supplementary Materials. It includes descriptions about the ground truth annotations provided by CAMI II challenges (Section A), a brief description of k-means++ method (Section B), descriptions about the parameters used in Step 4 of MetaBinner (Section C), Fig. S1 and Tables S1-S4, and commands for executing the compared binning methods and assemblers (Section F).

Additional file 2. Review history.

---

#### Peer review information

#### Review history
The review history is available as Additional file 2.

#### Authors' contributions
SZ conceived and supervised the project. SZ and ZW designed the study and the methodological framework. ZW and RY implemented the methods. PH and ZW carried out the computational analyses. ZW drafted the paper. FS and SZ modified the paper. FS, SZ, and ZW finalized the paper. All authors agree to the content of the final paper.

#### Funding
This work was supported by Shanghai Municipal Science and Technology Major Project [No.2018SHZDZX01], National Natural Science Foundation of China [No. 62272105], the 111 Project [No. B18015], ZJ Lab, and Shanghai Center for Brain Science and Brain-Inspired Technology.

### Availability of data and materials

Source codes for MetaBinner are freely available on GitHub under a GNU GPLv3 license (https://github.com/ziyewang/MetaBinner) [44] with the version used in the manuscript deposited in a DOI-assigning repository Zenodo (https://doi.org/10.5281/zenodo.5667457) [45]. All the datasets used are publicly available. The simulated datasets (CAMI mouse gut, CAMI Airways, and CAMI Gastrointestinal tract) created by CAMI II Challenge [18] are available from the CAMI portal at https://data.cami-challenge.org. All the simulated datasets are also downloadable from their respective DOIs (CAMI mouse gut:10.4126/FRL01-006421672; CAMI Airways, and CAMI Gastrointestinal tract: 10.4126/FRL01-006425518). The sequencing samples of the STEC dataset are available from https://www.ebi.ac.uk (Project: PRJEB1775) [36]. The assembled contigs of the STEC dataset are available from Zenodo (https://doi.org/10.5281/zenodo.7392537) [46].

## Declarations

### Ethics approval and consent to participate
Not applicable.

### Consent for publication
All authors have approved the manuscript for submission.

### Competing interests
The authors declare that they have no competing interests.

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

## 