## [Additional file 2. Review history. · Genome Biology]

Review History

First round of review

Reviewer 1

Were you able to assess all statistics in the manuscript, including the appropriateness of statistical tests used? No

Were you able to directly test the methods? No

Comments to author:

The authors report a new ensemble approach MetaBinner for binning contigs from metagenomic co-assembly to derive metagenome-assembled genomes (MAGs). Although there are numerous tools that have been developed for binning, it is still a challenging problem. Comparing to existing binning tools (individual or ensemble approaches), the new tool seemed to show superior performance resulting in better MAGs, as tested on simulated datasets and a real dataset. The authors have also done different experiments to show the contribution of the different considerations ("partial seed", different combinations of the features, etc) to the performance of the new approach, which is great.

1) the proposed approach uses some ideas, and in some cases functions of the existing tools. It is not always easy for me to see if it is similar ideas that were implemented, or when a function of an existing tool was used. For example, it seems that `Binning_refiner` was used to integrate the different component binning results. For this case, was `Binning_refiner` used directly?

2) MetaBinner uses a complicated way to estimate the number of clusters from k_0 (Figure A1) -- what was the rationale behind using k_1 and k_2 ?

3) the paper uses one sentence to describe how the coverage profile was derived (using MetaWrap's `binning.sh` script). It needs to be expanded with a little more details.

4) on page 15, it says that MegaHit was used to co-assemble the reads for all test cases, but Table A2 mentioned metaSpades for the STEC dataset.

5) no definition of "binning score" was given in the paper, although it was used for selecting bins (Fig. 3)

6) BMC3C's had very bad performance for the CAMI airways and CAMI Gastrointestinal tract, but looked ok for CAMI mouse gut (Table 1). Perhaps a double check of its performance is needed, and also maybe some discussions will be good.

7) how many component binning results were used exactly by MetaBinner?

A typo: contings (Fig. 3).

Table 6, I think there is no need to include seconds for the running time.

It is good to include some information about the simulated datasets: how many genomes were used for simulating the datasets, and how many true bins were expected in the main paper (e.g., in Table 1 or Table 7).

For the MetaBinner tool (on github):

1. Minor typos on GitHub page, inconsistent spelling of "MetaBinner" (MeatBinner & Metabinner v. MetaBinner)
2. A visual diagram might be nice to include on their GitHub and a little more detail in general.
3. More details about how to access/set paths to scripts, such as `gen_coverage_file.sh`, would be nice. The command in the example will not work otherwise.
4. When running through the tutorial, a "segmentation fault" issue inhibited the complete execution of MetaBinner. This may have something to do with memory allocation according to reports online. In particular, the error occurs on line 68 of the `run_metabinner.sh` script - this is where thread counts are articulated in the code. As said, we weren't able to run MetaBinner on tprovided test_data successfully

Reviewer 2

Were you able to assess all statistics in the manuscript, including the appropriateness of statistical tests used? Yes: I don't think that an additional statistical review is needed.

Were you able to directly test the methods? Yes

Comments to author:

In this submission, the authors consider the problem of metagenomic binning. In metagenomics, binning is a fundamental process that clusters reads or assembled contigs to individual genomes. The proposed method MetaBinner first generates various binning results using multiple types of features, initialization and combinations and then applies a greedy strategy to derive the final binning results.

Major comments:

A standard approach to estimate bin quality is to use CheckM that checks completeness and contamination of the bin using single-copy marker genes. While most binning tools (CONCOCT, MaxBin, VAMB, etc.) do not incorporate CheckM in their binning process, MetaBinner directly uses CheckM in the binning process (e.g., splitting bins of high contamination in Step 4, Page 12 and selecting high-quality final bins in Step 5, Page 12). Hence, MetaBinner is expected to derive less contamination and high completeness bins. My concern is that using CheckM again to evaluate the resulting bins again (e.g., in the real STEC dataset, Line 39-40, Page 13) would be biased toward MetaBinner compared to other binning tools. A better *ensemble* approach may be explored to avoid using CheckM to greedily select the highest-quality bin iteratively from a vast number of candidate bins.

MetaBinner considers the first, second, and third quartile values of the numbers of contigs containing each single-copy gene to obtain initial clustering centers (Lines 58-59, Page 11). I am curious why the authors do not use more single-copy genes (or at least those lead to a similar estimated bin number, with respect to the bin number estimated in Appendix Figure A1) to generate more candidate binning results based on the 'Partial Seed' strategy. The greedy strategy in Step 5 will benefit from more candidate bins from more binning results. Moreover, while it makes sense to split high contamination (>50%) bins in MetaBinner, the reason to include the high completeness condition (>70%) needs to be justified (Line 24-25 Page 12). How these parameters/thresholds on contamination and completeness are chosen to trigger the split?

Step 5 of the MetaBinner algorithm (Page 12) and Figure 3 (Page 13) seem very ad-hoc and confusing. Firstly, MetabinA already combines the coverage and composition information used in MetabinB and MetabinC, respectively (Line 19-20, Page 7, Line 12, Page 11). Why do the authors to use Bin_refiner to further generate combinations BinAB, BinAC, BinBC and BinABC? Secondly, there is no definition of "high bin score" (Line 36, Page 12) and there is no description on "how to estimate bin score for each bin from all results" (Figure 3, Page 13). Thirdly, if CheckM is again used in the greedy selection of high-quality bins from a vast number of candidates in each iteration, it is not surprising that the final binning results of MetaBinner show high completeness and low contamination because other baselines do not have the privilege to use CheckM to refine/adjust their outputs.

Minor comments:

Which taxonomic level is used for measuring the performances of tools using AMBER in Table 1 (Page 5) as well as getting the ground-true bin number in Table A1 (Page 15)? Is it at the species level or strain level?

How to prepare the ground truth annotations of contigs for AMBER (Table 1, Page 5)? Were the contigs mapped to the reference genomes?

Why do the authors change the value of k in increments/decrements of 5 in Figure A1 (Page 14)? Will there be an upper bound for k (for example, the maximum value of the numbers of contigs containing a single-copy marker gene)?

In Table A1 (page 25), results with "*" are from the CAMI tutorial. It would be better if the authors can run these tools using the latest dataset in this submission because the assembled contigs may not be identical when comparing those used in this submission to those used in the CAMI tutorial.

Is $D(x)$ be the shortest distance between point x and the closest previously chosen initial center (Line 60, Page 14)? (Reference: <http://ilpubs.stanford.edu:8090/778/1/2006-13.pdf>)

In the Step 3 (page 11), the description of the "Partial Seed" strategy looks very confusing in the first readthrough. I would appreciate if the authors can explain the meaning of K and l in a clear and concise way (even use diagrams if needed).

It would be better if the authors can include all exact commands used to run the assembly and binning tools so others can reproduce these results.

Reviewer #1:

> The authors report a new ensemble approach MetaBinner for binning contigs
> from metagenomic co-assembly to derive metagenome-assembled genomes (MAGs).
> Although there are numerous tools that have been developed for binning, it is
> still a challenging problem. Comparing to existing binning tools (individual
> or ensemble approaches), the new tool seemed to show superior performance
> resulting in better MAGs, as tested on simulated datasets and a real dataset.
> The authors have also done different experiments to show the contribution of
> the different considerations ("partial seed", different combinations of the
> features, etc) to the performance of the new approach, which is great.

Thanks for the comments.

> 1) the proposed approach uses some ideas, and in some cases functions of the
> existing tools. It is not always easy for me to see if it is similar ideas that
> were implemented, or when a function of an existing tool was used. For example,
> it seems that Binning_refiner was used to integrate the different component
> binning results. For this case, was Binning_refinner used directly?

In this paper, Binning_refiner was used directly to integrate the different binning results during the second stage of the ensemble step. We ran CheckM directly for one binning result and obtained the contigs having the single-copy genes in Step 4 (Step 4: Split bins with high contamination). Other binning-related tools were not used directly in MetaBinner. According to the reviewer's comments, we have added the Table 8 (Table R1 in this response letter) to introduce the differences between our ensemble strategy and other ensemble binners and added more descriptions in the Section "Step 5: Incorporate the component binning results with an ensemble module."

"Our two-stage strategy can integrate the twelve component binning results efficiently by avoiding running CheckM on multiple component binning results, which is quite time-consuming. The differences between our ensemble strategy and other ensemble binners are summarized in Table 8."

Furthermore, to avoid confusion, we have adjusted the writing order of the section "Step 2:..." and summarized our main contributions into the "Introduction" section as follows:

"The main contributions of MetaBinner are as follows: 1) MetaBinner uses a novel "partial seed" strategy for k-means initialization to utilize SCG information in the clustering process and obtain component binning results with high quality. 2) MetaBinner uses multiple different features and initializations for k-means clustering to obtain results with diversity for integration. 3) MetaBinner uses a novel effective, and efficient two-stage ensemble strategy based on MetaWRAP and UniteM to select the bins with high completeness and low contamination as the final results."

Table R1: The comparison between MetaBinner and other ensemble binners in ensemble strategy.

	BMC3C	DAS Tool	UniteM	MetaWRAP	MetaBinner
1. Independent of other individual binners	✓	×	×	×	✓
2. Use single-copy gene (SCG) information	×	✓	✓	✓	✓
3. Run CheckM on only one of all component binning results	–	–	×	×	✓
4. Use Binning_refiner to generate some of the candidate bins	×	×	×	✓	✓
5. Can directly integrate more than three component binning results	–	✓	✓	×	✓

“✓” denotes that the binner conforms to the statement given in the first column. “×” denotes that the binner does not conform to the statement given in the first column. “–” denotes “not applicable”.

> 2) MetaBinner uses a complicated way to estimate the number of clusters from k0
> (Figure A1) -- what was the rational behind using k1 and k2?

As described in the manuscript, the way to estimate the number of clusters is similar to that in SolidBin. As stated in SolidBin [1], the method finds two local maxima of the silhouette coefficient values, and the larger one corresponds to a bin number, which is the estimated bin number. We used k1 and k2 to avoid the process of estimating the number of clusters stopping early at the first local optimum.

> 3) the paper uses one sentence to describe how the coverage profile was derived
> (using MetaWrap’s binning.sh script). It needs to be expanded with a little
> more details.

In this revision, we followed the reviewer’s suggestion and added more details to describe how the coverage profile was derived into the Section “Implementation and parameter settings”:

“The coverage profiles of the contigs were obtained via a script of MetaWRAP 1.2.1, “binning.sh”, while running the MetaWRAP’s binning module. This script aligns the reads against the contigs using BWA [43] and calculates the mean of the base depths (coverage) for each contig.”

We also added the detailed commands of executing the compared binning methods into Additional file 2.

> 4) on page 15, it says that MegaHit was used to co-assemble the reads for all
> test cases, but Table A2 mentioned metaSpades for the STEC dataset.

We thank the reviewer for pointing it out. In this revision, we have marked which results came from which assembly in Table 2, Table 7, Additional file 1: Table S1 and Table S2. We also added the corresponding descriptions into the Section “The real dataset”:

“The binning results based on the MEGAHIT assembly are shown in the “Results” section. The quality of the assemblies may affect the follow-up binning performance. Therefore, we also used the MetaWRAP-Assembly module and the metaSPAdes assembler to co-assemble the reads from all the samples. The binning results based on this assembly are given in Additional file 1: Table S2.”

> 5) no definition of "binning score" was given in the paper, although it was
> used for selecting bins (Fig. 3)

Thanks for pointing it out. We have added the definition into the “Step 5:...” section and the descriptions of Figure 3 as follows:

“The quality of a bin is described by a bin score, which is computed as $100 \times (\text{completeness} - 3 \times \text{contamination})$.”

> 6) BMC3C's had very bad performance for the CAMI airways and CAMI
> Gastrointestinal tract, but looked ok for CAMI mouse gut (Table 1). Perhaps
> a double check of its performance is needed, and also maybe some discussions
> will be good.

We noticed that before and did a double-check. BMC3C is not widely used, and it also achieves very bad performance in a benchmark study published in 2020 [2]. BMC3C automatically estimates the number of clusters and initializes it based on the number of contigs, which affects the stability of its performance. As shown in Additional file 1: Table S1 and Table 7, the numbers of predicted bins of BMC3C are quite different from those of other bidders and the number of expected bins for the CAMI Airways dataset. For example, BMC3C generates 1560 bins for CAMI Airways, while the number of expected bins is 753. In this revision, we have added the corresponding descriptions into the Section “MetaBinner outperforms other available contig binning methods on the simulated datasets evaluated by AMBER”:

“BMC3C automatically estimates the number of clusters and initializes it based on the number of contigs, which affects the stability of its performance.”

“As shown in Additional file 1: Table S1 and Table 7, the numbers of predicted bins of BMC3C are quite different from those of other bidders and the number of expected bins for the CAMI Airways dataset. For example, BMC3C generates 1560 bins for CAMI Airways, while the number of expected bins is 753.”

> 7) how many component binning results were used exactly by MetaBinner?

Table R2: Datasets used in the experiments.

Data Type	Dataset	# samples	# contigs (> 1000bp)	# genomes	# expected bins
Simulated	CAMI Airways	10	285047	828	753
	CAMI Gastrointestinal tract	10	57088	259	246
	CAMI Mouse gut	64	241451	791	769
Real	STEC (MEGAHIT assembly)	53	255484
	STEC (metaSPAdes assembly)	53	226544

“# genomes” denotes the number of genomes used for simulating the datasets. “# expected bins” denotes the number of true bins expected among the contigs (> 1000bp).

There are 12 component binning results used exactly by MetaBinner. There are three feature matrices (X_{combo} , X_{cov} , and X_{com}) for each dataset. MetaBinner generated four binning results using each feature matrix. In this revision, we have added the corresponding descriptions into the Section “Step 3: Generate binning results with multiple features and initializations”:

“, twelve binning results are generated for next step.”

> A typo: `contings` (Fig. 3).

Sorry for the carelessness. We have corrected the typo.

> Table 6, I think there is no need to include seconds for the running time.

We followed the reviewer’s suggestion and removed the seconds for the running time in Table 6.

> It is good to include some information about the simulated datasets: how many genomes were used for simulating the datasets, and how many true bins were expected in the main paper (e.g., in Table 1 or Table 7).

We followed the reviewer’s suggestion and added the corresponding values and descriptions in Table 7, as shown in Table R2 in this response letter.

> For the MetaBinner tool (on github):

> 1. Minor typos on GitHub page, inconsistent spelling of "MetaBinner"
> (MeatBinner & Metabinner v. MetaBinner)

Sorry for the carelessness. We have corrected the inconsistent spelling.

> 2. A visual diagram might be nice to include on their GitHub and a little more
> detail in general.

We followed the reviewer's suggestion and added a visual diagram and more details on our GitHub.

> 3. More details about how to access/set paths to scripts, such as
> gen_coverage_file.sh, would be nice. The command in the example will not work
> otherwise.

We followed the reviewer's suggestion and added more details about how to access/set paths to scripts on our GitHub.

> 4. When running through the tutorial, a "segmentation fault" issue inhibited
> the complete execution of MetaBinner. This may have something to do with
> memory allocation according to reports online. In particular, the error occurs
> on line 68 of the run_metabinner.sh script - this is where thread counts
> are articulated in the code. As said, we weren't able to run MetaBinner on
> tprovided test_data successfully

It seems that there is something wrong while generating the component binning results. We are sorry that we do not know the exact reason why it could not run successfully, because there is no more complete log. Our test data contains 5889 contigs, and it requires about 3.3 GB (Memory) to run the codes for generating the component binning results on the test data. We have carried out tests on both the bioconda package and the GitHub version of MetaBinner using different servers. We also have asked other people to help to test MetaBinner. All of these tests run successfully. You can try MetaBinner with a different server that meets the memory requirement. If it doesn't work, please provide us with more details. We will try to solve the problem. In addition, we will provide continuous maintenance of the codebase and address the issues raised on GitHub.

Reviewer #2:

> In this submission, the authors consider the problem of metagenomic binning. In
> metagenomics, binning is a fundamental process that clusters reads or assembled
> contigs to individual genomes. The proposed method MetaBinner first generates
> various binning results using multiple types of features, initialization and
> combinations and then applies a greedy strategy to derive the final binning
> results.

Thanks for the comments.

Major comments:

> A standard approach to estimate bin quality is to use CheckM that checks
> completeness and contamination of the bin using single-copy marker genes. While
> most binning tools (CONCOCT, MaxBin, VAMB, etc.) do not incorporate CheckM in
> their binning process, MetaBinner directly uses CheckM in the binning process
> (e.g., splitting bins of high contamination in Step 4, Page 12 and selecting
> high-quality final bins in Step 5, Page 12). Hence, MetaBinner is expected
> to derive less contamination and high completeness bins. My concern is that
> using CheckM again to evaluate the resulting bins again (e.g., in the real
> STEC dataset, Line 39-40, Page 13) would be biased toward MetaBinner compared
> to other binning tools. A better *ensemble* approach may be explored to avoid
> using CheckM to greedily select the highest-quality bin iteratively from a vast
> number of candidate bins.

We thank the reviewer for this insight. We agree that it is possible that using CheckM again to evaluate the resulting bins for the real dataset again would be biased toward MetaBinner (using domain-specific marker sets of CheckM for estimating bin scores) and MetaWRAP (using CheckM directly). In this revision, we tried another way to evaluate the binning performance for the real dataset. We obtained the species-level annotations by aligning the contigs to the NCBI's nt database, and further used AMBER [3] to evaluate the binner based on the contigs labeled on the species level. Because there is no ground truth for the real dataset, there is no best way to evaluate the binning performance using the STEC dataset. This evaluation method, which evaluates the binning performance based on the annotated contigs, can be used as a reference. The results are shown in Table R3 in this response letter.

In this revision, we added the corresponding descriptions into the Section "MetaBinner produces the most near-complete MAGs on the real dataset":

"It is possible that using CheckM to evaluate the resulting bins for the real dataset would be biased toward MetaBinner (using domain-specific marker sets of CheckM for estimating bin scores) and MetaWRAP (using CheckM directly). Therefore, we explored another approach to evaluate the binning

Table R3: Performance comparison of the binners on the real dataset evaluated by AMBER (based on the contigs labeled on the species-level).

Dataset	Methods	Metrics					
		#bins (>50% comp <10% cont)	#bins (>70% comp <10% cont)	#bins (>90% comp <10% cont)	#bins (>50% comp <5% cont)	#bins (>70% comp <5% cont)	#bins (>90% comp <5% cont)
STEC (MEGAHIT assembly)	CONCOCT	51	45	18	46	41	17
	MaxBin	47	40	24	37	31	19
	MetaBAT	47	24	6	44	22	5
	VAMB	57	47	20	50	42	19
	BMC3C	61	43	21	54	39	18
	DAS Tool	37	32	14	31	28	12
	MetaWRAP	53	39	11	49	36	10
	MetaBinner	50	44	25	46	41	24

The best results among all the methods are in bold. The input binning results of MetaWRAP and DAS Tool are generated by CONCOCT, MaxBin and MetaBAT. “#bins (>50% comp <10% cont)” denotes that the number of recovered bins that have >50% completeness and <10% contamination.

performance for the real dataset. First, we obtained the species-level annotations of the contigs by aligning them to the NCBI’s nt database using TAXAassign v0.4 (<https://github.com/umerijaz/taxaassign>) as done in [11, 34], with the default parameters. For the STEC (MEGAHIT assembly) dataset, a total of 46 609 out of 255 484 contigs were labeled on the species level for evaluation. Then, we used AMBER [27] to evaluate the binners based on the contigs labeled on the species level. Note that we did not re-run the binning methods but extracted labeled contigs from the results of each binner for evaluation. As shown in Additional file 1: Table S3, MetaBinner still performs the best in producing the most MAGs with the highest completeness (>90% completeness and <5% or <10% contamination). For example, MetaBinner produces 24 most near-complete MAGs (>90% completeness and <5% contamination), which is followed by VAMB and MaxBin (19 near-complete MAGs). We also find that VAMB and BMC3C perform well in producing middle-complete and less-complete MAGs (>70% or >50% completeness). These suggest that VAMB and BMC3C generate bins with low contamination but low completeness. As shown in Additional file 1: Table S1, the total number of predicted bins of BMC3C for STEC (MEGAHIT assembly) is much larger than other binners, which means that BMC3C may produce more bins with low contamination and low completeness. On the other hand, VAMB clustered the contigs into 75 526 bins first and then kept the bins more than 200 000bp as the predicted bins. This means that VAMB tends to generate bins with high purity but less completeness, which is consistent with the result of CAMI II [18]. All these results highlight the advantage of MetaBinner in producing the most near-complete MAGs on the real dataset.”

Considering the reviewer’s suggestion, we will try to explore other ensemble approaches without using CheckM in the future.

> MetaBinner considers the first, second, and third quartile values of the
> numbers of contigs containing each single-copy gene to obtain initial
> clustering centers (Lines 58-59, Page 11). I am curious why the authors do
> not use more single-copy genes (or at least those lead to a similar estimated
> bin number, with respect to the bin number estimated in Appendix Figure A1) to
> generate more candidate binning results based on the 'Partial Seed' strategy.
> The greedy strategy in Step 5 will benefit from more candidate bins from more
> binning results. Moreover, while it makes sense to split high contamination
> (>50%) bins in MetaBinner, the reason to include the high completeness
> condition (>70%) needs to be justified (Line 24-25 Page 12). How these
> parameters/thresholds on contamination and completeness are chosen to trigger
> the split?

It will take more time for generating the component binning results if we use more single-copy genes. We cannot guarantee that the estimated bin number is very close to the actual number of species in metagenomes. Therefore, we did not use the single-copy genes those lead to a similar estimated bin number, with respect to the bin number estimated in Additional file 1: Figure S1. The first, second, and third quartile values of the numbers of contigs expand the selection range, which can reduce the influence of the limitation of estimated bin number.

According to the definitions for estimated completeness and contamination given in CheckM, it is possible for a bin with high contamination (>50%) and low completeness (<50%). Therefore, we set a high completeness condition to split bins in MetaBinner to avoid unnecessary waste of time on the bins with low completeness. If we set the high completeness condition as >50%, it will be easier to fail to meet the requirements of a high-quality bin (completeness (>50%) and contamination (<10%)) after splitting. Therefore, we set this high completeness condition (>70%). CheckM estimates contamination based on the number of multi-copy marker genes identified, and the contamination may exceed 100%. We set these parameters to make many high contamination bins be split but reduce unnecessary waste of time. We further tried different values of parameters for MetaBinner in CAMI Airways, as shown in Additional file 1: Table S4 and Table R4 in this response letter.

In this revision, we have added the corresponding descriptions into Additional file 1:

“Step 4 is developed for generating more low-contamination bins for further integration by splitting bin with high contamination. According to the definitions for estimated completeness and contamination given in CheckM, it is possible for a bin with high contamination (>50%) and low completeness (<50%). Therefore, we also set a high completeness condition to split bins to avoid unnecessary waste of time on the bins with low completeness. We further tried different values of parameters for MetaBinner in CAMI Airways, as shown in Additional file 1: Table S4.”

Table R4: Performance comparison of the binners on the CAMI Airways dataset evaluated by AMBER.

Dataset	Methods	Metrics					
		#bins (>50% comp <10% cont)	#bins (>70% comp <10% cont)	#bins (>90% comp <10% cont)	#bins (>50% comp <5% cont)	#bins (>70% comp <5% cont)	#bins (>90% comp <5% cont)
CAMI Airways	MetaBinner (post_process: mincomp_50_mincont_10)	217	196	140	186	172	125
	MetaBinner (post_process: mincomp_50_mincont_30)	215	191	142	186	169	127
	MetaBinner (post_process: mincomp_50_mincont_50)	215	191	144	186	169	129
	MetaBinner (post_process: mincomp_70_mincont_10)	217	196	140	186	172	125
	MetaBinner (post_process: mincomp_70_mincont_30)	215	191	142	186	169	127
	MetaBinner (post_process: mincomp_70_mincont_50)	215	191	144	186	169	129

“#bins (>50% comp <10% cont)” denotes the number of recovered bins that have >50% completeness and <10% contamination. “mincomp_a_mincont_b” denotes that we split bins with the contamination ($\geq b\%$) and completeness ($\geq a\%$) in Step 4.

> Step 5 of the MetaBinner algorithm (Page 12) and Figure 3 (Page 13) seem very
> ad-hoc and confusing. Firstly, MetabinA already combines the coverage and
> composition information used in MetabinB and MetabinC, respectively (Line
> 19-20, Page 7, Line 12, Page 11). Why do the authors to use Bin_refiner to
> further generate combinations BinAB, BinAC, BinBC and BinABC? Secondly,
> there is no definition of "high bin score" (Line 36, Page 12) and there is
> no description on "how to estimate bin score for each bin from all results"
> (Figure 3, Page 13). Thirdly, if CheckM is again used in the greedy selection
> of high-quality bins from a vast number of candidates in each iteration, it
> is not surprising that the final binning results of MetaBinner show high
> completeness and low contamination because other baselines do not have the
> privilege to use CheckM to refine/adjust their outputs.

Binning_refiner can reduce the contamination level of the component binning results by refining bins according to the shared contigs between different binning results, but at the cost of low completeness. It can be used as an intermediate step to produce more low-contamination bins for further selection.

We have added the definition into the descriptions of Figure 3 as follows:

“The quality of a bin is described by a bin score, which is computed as $100 \times (\text{completeness} - 3 \times$

contamination). If a bin has a bin score higher than 10, low contamination (<15%) and high completeness (>50%), we regard it as a bin with high score.”

We explored a different approach to evaluate the binning performance on the real dataset and to explain the question about CheckM in the response to the reviewer’s first question.

Minor comments:

> Which taxonomic level is used for measuring the performances of tools using
> AMBER in Table 1 (Page 5) as well as getting the ground-true bin number in
> Table A1 (Page 15)? Is it at the species level or strain level?

Strain-level is used for measuring the performances of tools using AMBER. Take the CAMI mouse gut as an example. The simulated dataset was generated with CAMISIM [4] from 791 public prokaryotic genomes, which comprise 549 species [5]. We have included the number of genomes used for simulating the datasets and the number of actual bins expected in Table 7 (Table R2 in this response letter).

We have added the related descriptions in the Additional file 1: Section A: “Descriptions about the ground truth annotations provided by CAMI II challenges”:

“Strain-level is used for measuring the performances of tools using AMBER.”

> How to prepare the ground truth annotations of contigs for AMBER (Table 1, Page
> 5)? Were the contigs mapped to the reference genomes?

The ground truth annotations of contigs for AMBER are provided by the organizer of CAMI I and II challenges (<https://data.cami-challenge.org>). The simulated datasets were generated with CAMISIM [4]. As introduced in [4, 6, 7], CAMISIM generates a BAM file for each sample of a dataset, which gives the alignment of simulated reads to reference genomes. Furthermore, it extracts the perfect co-assembly of all samples by including all regions covered by at least one read according to the BAM files. The perfect co-assembly is used for binning, named as “gold-standard cross-sample assembly” in our paper, as done in [5].

We have added more descriptions about the ground truth annotations of contigs in the Additional file 1: Section A: “Descriptions about the ground truth annotations provided by CAMI II challenges”:

“The ground truth annotations of contigs from the simulated datasets for AMBER are provided by the organizer of CAMI I and II challenges (<https://data.cami-challenge.org>). The simulated datasets were generated with CAMISIM [1]. As introduced in [1-3], CAMISIM generates a BAM file for each sample of a dataset, which gives the alignment of simulated reads to reference genomes. Furthermore, it extracts the perfect co-assembly of all samples by including all regions covered by at least one read according to the BAM files. The perfect co-assembly is used for binning, named as “gold-standard cross-sample assembly” in our paper, as done in [4].”

> Why do the authors change the value of k in increments/decrements of 5 in
> Figure A1 (Page 14)? Will there be an upper bound for k (for example, the
> maximum value of the numbers of contigs containing a single-copy marker gene)?

It is a trade-off between the time cost and performance. For a much larger step such as 10 or 20, it will be easy to miss a better k and harm the performance. While for a much smaller step such as 1 or 2, it may try too many times and waste time.

In our codes, we set an upper bound for k using $\min(3 \times (k_0 + 1), \text{the number of contigs})$. Using the maximum value of the numbers of contigs containing a single-copy marker gene is a good idea. Actually, according to the reviewer's comments, we checked the estimated bin number and the maximum value of the numbers of contigs containing a single-copy marker gene for each simulated dataset. The estimated bin numbers are much lower than the maximum values of the numbers of contigs containing a single-copy marker gene and the upper bound for k we set.

> In Table A1 (page 25), results with "*" are from the CAMI tutorial. It would
> be better if the authors can run these tools using the latest dataset in this
> submission because the assembled contigs may not be identical when comparing
> those used in this submission to those used in the CAMI tutorial.

We followed the reviewer's suggestion and made corresponding changes in Table 1, Additional file 1: Table S1 and related texts.

> Is $D(x)$ be the shortest distance between point x and the closest
> previously chosen initial center (Line 60, Page 14)? (Reference:
> <http://ilpubs.stanford.edu:8090/778/1/2006-13.pdf>)

Yes, that is what we mean. To make the description clearer, we modified the descriptions as follows: "where $D(x)$ is the shortest distance between point x and the closest previously chosen initial center."

> In the Step 3 (page 11), the description of the "Partial Seed" strategy looks
> very confusing in the first readthrough. I would appreciate if the authors can
> explain the meaning of K and l in a clear and concise way (even use diagrams if
> needed).

We followed the reviewer's suggestion and rewrote the related description in the Section "Step 3: Generate binning results with multiple features and initializations" as follows:

"Let K denote the bin number estimated by Step 2. Instead of randomly choosing the K cluster

centers, we used SCG information to define cluster centers. For a particular SCG, suppose that there are l ($l < K$) contigs containing this gene. These l contigs should belong to different genomes, which were used to initialize l cluster centers. Then we used k-means++ to generate the other $K - l$ initial clustering centers and get the binning results.”

> It would be better if the authors can include all exact commands used to run
> the assembly and binning tools so others can reproduce these results.

We followed the reviewer’s suggestion and added the commands used to run the assembly and binning tools into the Additional file 2. We further added the corresponding descriptions into the Section “Implementation and parameter settings” as follows:

“The detailed commands of executing the compared binning methods and assemblers are given in Additional file 2.”

References

- [1] Wang, Z., Wang, Z., Lu, Y.Y., Sun, F., Zhu, S.: SolidBin: improving metagenome binning with semi-supervised normalized cut. *Bioinformatics* **35**(21), 4229–4238 (2019)
- [2] Yue, Y., Huang, H., Qi, Z., Dou, H.M., Liu, X.Y., Han, T.F., Chen, Y., Song, X.J., Zhang, Y.H., Tu, J.: Evaluating metagenomics tools for genome binning with real metagenomic datasets and CAMI datasets. *BMC Bioinformatics* **21**(1), 334 (2020)
- [3] Meyer, F., Hofmann, P., Belmann, P., Garrido-Oter, R., Fritz, A., Sczyrba, A., McHardy, A.C.: AMBER: Assessment of Metagenome BinnERS. *Gigascience* **7**(6) (2018)
- [4] Fritz, A., Hofmann, P., Majda, S., Dahms, E., Dröge, J., Fiedler, J., Lesker, T.R., Belmann, P., DeMaere, M.Z., Darling, A.E., Sczyrba, A., Bremges, A., McHardy, A.C.: CAMISIM: simulating metagenomes and microbial communities. *Microbiome* **7**(1), 17 (2019)
- [5] Meyer, F., Lesker, T.R., Koslicki, D., Fritz, A., Gurevich, A., Darling, A.E., Sczyrba, A., Bremges, A., McHardy, A.C.: Tutorial: assessing metagenomics software with the CAMI benchmarking toolkit. *Nat. Protoc.* **16**(4), 1785–1801 (2021)
- [6] Sczyrba, A., Hofmann, P., Belmann, P., Koslicki, D., Janssen, S., Dröge, J., *et al.*: Critical Assessment of Metagenome Interpretation—a benchmark of metagenomics software. *Nat. Methods* **14**(11), 1063–1071 (2017)
- [7] Meyer, F., Fritz, A., Deng, Z.-L., Koslicki, D., Gurevich, A., *et al.*: Critical assessment of metagenome interpretation - the second round of challenges. *bioRxiv* (2021). doi:10.1101/2021.07.12.451567

Second round of review

Reviewer 2

Thanks for the time and effort on addressing my comments.

While Table 2 (CheckM evaluation) demonstrates that MetaBinner clearly outperforms other baselines, the performance of MetaBinner does not stand out in Table R3 (AMBER evaluation) on the same dataset. Therefore, I am still concerned that CheckM is used to evaluate performance whilst also apparently being part of the MetaBinner pipeline, which would bias the evaluation in Table 2. More discussions and analysis on the discrepancy between Table 2 and Table R3 would be much appreciated.

Authors Response

Point-by-point responses to the reviewers' comments:

Thanks for the time and effort on addressing my comments.

While Table 2 (CheckM evaluation) demonstrates that MetaBinner clearly outperforms other baselines, the performance of MetaBinner does not stand out in Table R3 (AMBER evaluation) on the same dataset. Therefore, I am still concerned that CheckM is used to evaluate performance whilst also apparently being part of the MetaBinner pipeline, which would bias the evaluation in Table 2. More discussions and analysis on the discrepancy between Table 2 and Table R3 would be much appreciated.

Response

We have added the following explanation and analysis on the discrepancy between Table 2 and Table S3 into the "Discussion" Section based on the comments of Referee 2.

"MetaBinner achieves the best overall performance on the real dataset (Tables 2 and S3), especially using CheckM for evaluation. On the other hand, VAMB and BMC3C performed pretty well based on AMBER evaluation for the contigs with species-level annotation (Table S3). Here we propose a possible explanation. Only the annotated contigs can be used for AMBER evaluation, but the annotated contigs of each species cannot cover the entire genome. Therefore, the evaluation methods may be biased towards the bidders that generate bins with low contamination but low completeness. To prove our hypothesis to a certain extent, we regarded the annotated contigs of each species as a putative bin. We then evaluated the contamination and completeness of these putative bins using CheckM. Among the 256 putative bins, 252 bins have 50% completeness. It is still challenging to evaluate the contamination and completeness of the genomes from the real metagenomes without bias."